



# Cold air outbreaks drive near-surface baroclinicity variability

Andrea Marcheggiani[1] and Thomas Spengler[1]

[1]Geophysical Institute, University of Bergen, and Bjerknes Centre for Climate Research, Bergen, Norway

**Correspondence:** Andrea Marcheggiani (andrea.marcheggiani@uib.no)

**Abstract.**

Cold air outbreaks (CAOs) are key drivers of near-surface baroclinicity in midlatitude oceanic regions, where cold continental air masses interact with warm sea surface temperatures, giving rise to strong surface heat fluxes. Despite their relatively limited spatio-temporal extent, CAOs exert a disproportionate influence on the variability of near-surface baroclinicity, particularly in the entrance regions of the North Atlantic and North Pacific storm tracks. To further clarify this relationship, we use the isentropic slope framework to distinguish between diabatic and adiabatic changes in baroclinicity and quantify the contribution of CAOs to near-surface baroclinicity variability in the Gulf Stream and Kuroshio-Oyashio extension regions.

Moderate-intensity CAOs account for up to 40% of the total near-surface baroclinicity variability in the Gulf Stream Extension, while occupying less than 15% of the region. In the Kuroshio-Oyashio Extension, CAOs explain a smaller fraction of variability despite their broader spatial extent. We employ phase space analysis to diagnose the typical phasing between adiabatic depletion and diabatic restoration of baroclinicity, with the former leading in time on the latter. Phase portraits and synoptic composites focused on CAO-related variability show that this characteristic phasing is predominantly linked to CAOs, whereas background variability contributes weakly and incoherently. These findings highlight the central role of CAOs in shaping near-surface baroclinicity and suggest that they are essential to the evolution of midlatitude storm tracks.

## 1 Introduction

Marine cold air outbreaks (CAOs) are associated with the discharge of cold polar air masses over relatively warm waters, giving rise to intense surface heat fluxes that dominate the total surface heat exchange over polar and subpolar oceans (e.g., Jensen et al., 2011; Isachsen et al., 2013; Harden et al., 2015; Papritz et al., 2015; Papritz and Spengler, 2017). While the occurrence and frequency of CAOs are primarily governed by the large-scale atmospheric flow (Kolstad et al., 2009; Iwasaki et al., 2014; Papritz and Spengler, 2017), the enhancement of temperature gradients during CAOs provides additional baroclinicity variability, potentially feeding back onto the large-scale flow (Charney, 1947; Eady, 1949; Hoskins and Valdes, 1990; Papritz and Spengler, 2015). This two-way interaction is particularly relevant in midlatitude storm tracks, where baroclinicity in the near-surface troposphere is typically depleted adiabatically and subsequently restored diabatically (Marcheggiani and Spengler, 2023). While synoptic features, such as cyclones, fronts, and atmospheric rivers are known to drive baroclinicity variability aloft (Marcheggiani et al., 2025), the specific role of CAOs in modulating near-surface baroclinicity has not been clarified. We



complement the upper tropospheric results of Marcheggiani et al. (2025) by quantifying the relative contribution of CAOs to near-surface baroclinicity variability in the entrance regions of the North Atlantic and North Pacific storm tracks.

Beyond their thermodynamic effects, CAOs exert a broader influence on the atmospheric circulation, promoting the formation of polar lows (Terpstra et al., 2016; Michel et al., 2018; Terpstra et al., 2021), moistening air masses that can lead to atmospheric blocking (Wenta et al., 2024), and contributing to the total meridional heat transport (Messori and Czaja, 2013; Pithan et al., 2018). CAOs are most frequent and intense near the sea ice margin and over oceanic western boundary currents, such as the Gulf Stream and the Kuroshio-Oyashio currents (Kolstad et al., 2009; Fletcher et al., 2016). These regions are characterised by strong sea surface temperature (SST) gradients, which amplify the air-sea temperature contrast during CAOs and yield diabatic restoration of near-surface baroclinicity (Nakamura et al., 2008; Sampe et al., 2010; Papritz and Spengler, 2015). However, the extent to which this interaction modulates baroclinicity variability, and thereby storm track dynamics, remains an open question.

To gain a clearer understanding of how CAOs modulate storm track dynamics, one needs to disentangle the mechanisms that drive baroclinicity variability in lower troposphere. While latent heating is the dominant source of diabatic baroclinicity production in the upper troposphere (Papritz and Spengler, 2015; Weijenborg and Spengler, 2020; Marcheggiani and Spengler, 2023), surface sensible heat fluxes play a prominent role in the lower troposphere(Nakamura et al., 2008; Hotta and Nakamura, 2011; Papritz and Spengler, 2015; Marcheggiani and Ambaum, 2020; Marcheggiani and Spengler, 2023). This vertical dichotomy aligns with observed differences in the phasing between adiabatic depletion and diabatic restoration of baroclinicity, whereby diabatic production associated with latent heating typically precedes adiabatic depletion of baroclinicity in the free troposphere (750–350 hPa)(Papritz and Spengler, 2015; Weijenborg and Spengler, 2020; Marcheggiani and Spengler, 2023), while baroclinicity near the surface (900–825 hPa) is restored diabatically following an initial adiabatic depletion, often associated with CAOs (Marcheggiani and Spengler, 2023). Given this dominant role of CAOs, we use the isentropic slope framework (Papritz and Spengler, 2015) to quantify their contribution to near-surface baroclinicity variability and depict the associated synoptic evolution.

## 2  Data and methods

We use data from the ERA5 reanalysis (Hersbach et al., 2020), interpolated onto a $0.5° \times 0.5°$ longitude-latitude grid. We use both 3-hourly instantaneous fields, as well as 6-hourly accumulated temperature tendencies due to parameterisations centred on each 3-hourly time step.

To compute the isentropic slope and its tendencies (Papritz and Spengler, 2015), we use temperature, geopotential height ($z$), wind velocity (u,v,w), and accumulated temperature tendencies due to parameterisations ($mttpm$) on 24 pressure levels (every 25 hPa between 1000 hPa and 750 hPa, every 50 hPa thereafter up to 100 hPa). Temperature tendencies are only available on model levels and were interpolated to the aforementioned pressure levels.

In the Northern Hemisphere, CAOs are most frequent and intense during the winter months, especially off the sea ice edge and over western sectors of the North Atlantic and North Pacific oceans (Kolstad et al., 2009; Kolstad, 2011; Fletcher et al.,



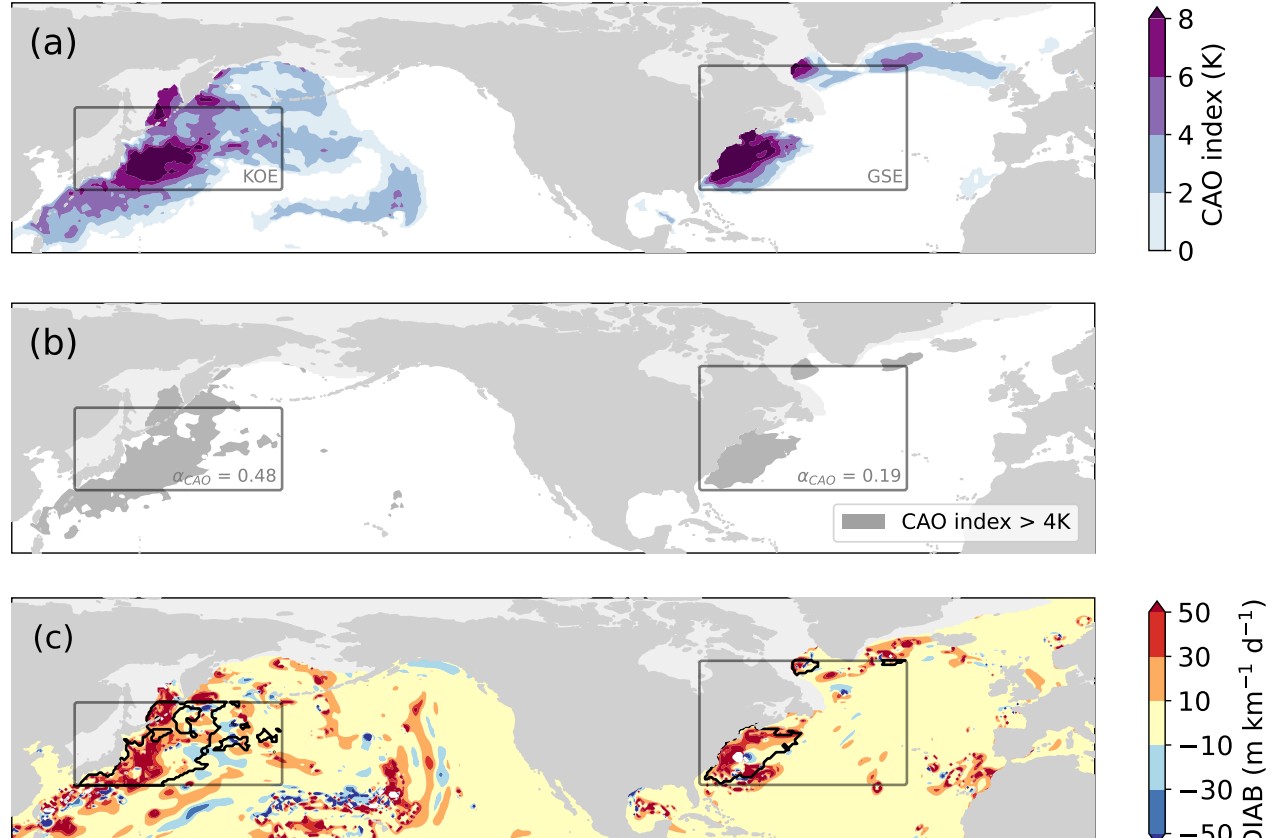

**Figure 1.** Graphical example of spatial averaging conditioned on CAOs for 23 January 2014 at 15Z. (a) CAO index (colour shading). (b) Mask $M$ for spatial averaging where the CAO index is above 4K (grey shading) within the GSE and KOE spatial domains (grey boxes). (c) DIAB, with CAO-only area-mean DIAB indicated by thick black contour. Land grid points, ocean grid points where sea ice concentration is above 15% for more than 5% of the time, and grid points pertaining to the Sea of Japan are masked out by grey shading.

2016). Hence we focus on extended winter months (November to February) from November 1979 to February 2020 over two
specific regions previously considered by Marcheggiani and Spengler (2023), namely the Gulf Stream Extension (GSE, 30°-60°N, 80°-30°W; Fig. 1) and Kuroshio-Oyashio Extension (KOE, 30°-50°N, 130°-180°E) regions. When calculating spatial averages, we exclude the Sea of Japan from the KOE region, as baroclinicity variability there is not necessarily linked to variability within the main North Pacific storm track. Furthermore, we also exclude grid points covered by land and where sea ice concentration is above 15% for more than 0.5% of the time to avoid misrepresentations of surface heat fluxes over sea ice
(Renfrew et al., 2021). We apply the resulting mask for all time steps to keep the total area constant.





## 2.1 Isentropic slope framework

We use the isentropic slope framework introduced by Papritz and Spengler (2015) to calculate diabatic and adiabatic tendencies in baroclinicity. Baroclinicity is measured as the slope $S \equiv |\nabla_\theta z|$ of isentropic surfaces, where $z$ is altitude and the subscript indicates a horizontal gradient on an isentropic surface. The Eulerian form of the slope tendency equation on pressure levels is

$$\left.\frac{\partial S}{\partial t}\right|_p = \underbrace{\frac{\nabla_\theta z}{S} \cdot \nabla_\theta w_{\mathrm{id}}}_{\mathrm{TILT}} \underbrace{-\frac{\partial z}{\partial \theta}\frac{\nabla_\theta z}{S} \cdot \nabla_\theta \dot{\theta}}_{\mathrm{DIAB}} \underbrace{-\frac{\partial S}{\partial \theta}\mathbf{v} \cdot \nabla \theta}_{\mathrm{ADV}}, \qquad (1)$$

where TILT represents adiabatic tilting by the isentropic displacement vertical wind $w_{\mathrm{id}}$ (Hoskins et al., 2003, their Equation 8), DIAB describes diabatic deformation due to heating, and ADV accounts for slope advection by the three-dimensional wind $\mathbf{v} = (u, v, \omega)$. The TILT and DIAB terms generally exert opposing effects on the slope (Papritz and Spengler, 2015; Marcheggiani and Spengler, 2023). ADV also represents adiabatic changes in slope and, especially near the surface, its magnitude is
comparable to the other two terms (Papritz and Spengler, 2015; Marcheggiani and Spengler, 2023). However, its interpretation in modifying the slope is less straight-forward compared to TILT and DIAB, thus we neglect it in this study.

As in Marcheggiani et al. (2025), data are first spatially filtered by spectral truncation to T84 to filter out small-scale features (e.g., gravity waves) that can be associated with extremely steep slopes that are not relevant to study larger scale baroclinic development. We focus on the near-surface troposphere and thus consider vertical averages of slope, TILT, and DIAB between
900hPa and 825hPa (consistent with Marcheggiani and Spengler, 2023).

## 2.2 Detection of Cold Air Outbreaks

To detect CAOs we follow (Papritz et al., 2015) and first calculate the CAO index, defined as air-sea potential temperature difference,

$$\mathrm{CAO\ index} = \theta_{\mathrm{sst}} - \theta_{850}, \qquad (2)$$

where $\theta_{\mathrm{sst}}$ is potential sea surface temperature and $\theta_{850}$ is potential temperature at 850hPa. CAOs are detected over those ocean
grid points where the CAO index exceeds a certain threshold, $\theta_T$. Different thresholds will result in the detection of CAOs of different intensity, with moderate intensity and stronger mostly captured by using $\theta_T = 4K$ (Papritz et al., 2015; Papritz, 2017; Papritz and Spengler, 2017).

## 2.3 Partitioning contributions from CAOs to baroclinicity variability

We study the temporal variability associated with the evolution of the North Atlantic and North Pacific storm tracks by con-
sidering spatial averages of slope, DIAB, and TILT (Eq. 1) over the GSE and KOE regions. Total variability is partitioned into CAO and non-CAO contributions following a similar approach to Marcheggiani et al. (2025). We first construct a CAO mask





$M$ based on the CAO index,

$$M_i(t) = M(\text{lon}_i, \text{lat}_i; t) = \begin{cases} 1 & \text{if CAO index} \geq \theta_T \\ 0 & \text{otherwise,} \end{cases} \tag{3}$$

which is then used to construct time series of CAO area mean,

$$\chi_{\text{CAO}}(t) = \frac{\sum_i a_i M_i(t) \chi_i(t)}{\sum_i a_i M_i(t)}, \tag{4}$$

and background, non-CAO area mean,

$$\chi_{\text{noCAO}}(t) = \frac{\sum_i a_i [1 - M_i(t)] \chi_i(t)}{\sum_i a_i [1 - M_i(t)]}. \tag{5}$$

Sums in Eqs. 4-5 are over all grid points in the averaging box. The areal extent of each grid point is represented by $a_i$, while $\chi_i$ represents the value of either slope, DIAB, or TILT at that grid point.

At each time step, CAOs occupy a fraction of the total area,

$$\alpha_{\text{CAO}}(t) = \frac{\sum_i a_i M_i(t)}{\sum_i a_i}. \tag{6}$$

By construction, the time series of the total area mean,

$$\chi_{\text{tot}} = \frac{\sum_i a_i \chi_i(t)}{\sum_i a_i}, \tag{7}$$

is the sum of the CAO component (Eq. 4) and non-CAO, background component (Eq. 5) weighted by the area fraction (Eq. 6) occupied by CAOs,

$$\chi_{\text{tot}} = \frac{\sum_i a_i \chi_i(t)}{\sum_i a_i} = \alpha \chi_{\text{CAO}} + (1 - \alpha) \chi_{\text{noCAO}}. \tag{8}$$

An example of the partition in CAO and background components is presented in Fig. 1. The CAO index is significantly larger than zero over large areas both in the North Atlantic and especially in the North Pacific oceans (Fig. 1a). Then we only consider CAOs of at least moderate intensity by choosing $\theta_T = 4$K and obtain the corresponding mask $M$ (Fig. 1b). Finally, we look at $\chi =$DIAB and calculate DIAB$_{\text{CAO}}$ as the area-mean over those grid points within the averaging domain that also belong to the mask, that is both inside the grey boxes and within the thick black contour (Fig. 1c). In this case, the CAO in the KOE region extends over almost half the domain ($\alpha_{\text{CAO}} = 0.48$), while only covering about a fifth in the GSE region ($\alpha_{\text{CAO}} = 0.19$).

## 2.4 Constructing the DIAB-TILT phase space

Using time series for area-mean DIAB and TILT defined in either Eq. 4, Eq. 5, or Eq. 8, we construct phase portraits for CAO, background, and total variability, respectively. Similar to Marcheggiani and Spengler (2023), we utilise a Gaussian kernel with averaging length scale equal to half of the standard deviation in either of the time series. We then define a stream function $\psi$ (black contours in Fig. 3) to visualise the non-divergent flow in the phase space, so that

$$\mathbf{F} = (\varrho \, c_x, \varrho \, c_y) = \left( -\frac{d\psi}{dy}, \frac{d\psi}{dx} \right), \tag{9}$$





where $\mathbf{F} = \varrho\mathbf{c}$ is the mass flux, $\varrho$ is the data density at each point in the phase space, and $\mathbf{c} = (c_x, c_y)$ represents the phase
space velocity field. We refer the reader to Novak et al. (2017) and Marcheggiani et al. (2022) for more technical details on the
construction of the phase portraits.

## 3 Quantifying CAO contribution to total baroclinicity variability

The strongest DIAB and TILT are generally collocated with the largest values of the CAO index (compare Fig. 1a and c). The
fraction of total variance explained by CAOs can be determined by the covariance between the total and the CAO time series,
normalised by the total variance,

$$R_{\text{CAO}} = \frac{\text{Cov}(\chi_{\text{tot}}, \chi_{\text{CAO}})}{\text{Var}(\chi_{\text{tot}})} \,, \tag{10}$$

where a high degree of similarity between two time series results in a ratio approaching one, suggesting that the variability in
one time series significantly contributes to the variability in the other. Conversely, if the two time series are largely uncorrelated,
or have significantly different variances, the ratio remains near zero, indicating a minimal contribution of the variability in one
time series to the other.
We assess the contribution of CAOs to total variability relative to their intensity as measured by the CAO index (Eq. 3).
Using larger CAO index thresholds ($\theta_T$ in Eq. 3) yields smaller CAO masks, evident from the decreasing area fractions in
Fig. 2 with increasing CAO index threshold.
   Over the GSE region, moderate and strong CAOs (CAO index$\geq$ 4K, Fig. 2) contribute disproportionately to the total variance
compared to their areal extent, with the strongest CAOs (CAO index$\geq$ 8K) extending over a minimal fraction of the domain
($\alpha_{\text{CAO}} \approx 0.04$) while explaining up to 5 times the total variance ($R_{\text{CAO}} \approx 0.2$). Thus CAOs explain almost half (40-45%) of
the total variance in baroclinicity in the GSE despite only extending over less than 20% of region. Including weaker CAOs
(CAO index$\geq$ 2K or $\geq$ 0K), the areal extent increases significantly (25%-50% of the domain) while the associated baroclinicity
variability explains up to 70% of the total variance. CAOs explain less variance in TILT compared to DIAB. This is consistent
with TILT being more noisy than DIAB, therefore more variability occurs in the entire averaging domain, also outside of CAO
masks.
   Over the KOE region, CAOs typically extend over larger areas compared to their counterparts in the GSE. This is evident
in the example in Fig. 1, where positive CAO-indices extend significantly farther south- and eastwards, away from the Asian
continent. The larger extent of CAOs over the KOE region can be ascribed in part to the East Asian winter monsoon (Zhang
et al., 1997; Compo et al., 1999), leading to a stronger cold air mass stream over the North Pacific compared to the North
Atlantic (Iwasaki et al., 2014).
   Moderate CAOs (CAO index$\geq$ 4K) extend over more than 20% of the KOE, compared to about 15% for GSE (Fig. 2b).
Although stronger CAOs (CAO index$\geq$ 6K) have comparable spatial extent in both regions ($\overline{\alpha}_{\text{CAO}} \approx 0.08$ in the KOE, $\overline{\alpha}_{\text{CAO}} \approx$
0.09 in the GSE), they account for a smaller share of the total variance in both TILT and DIAB in the KOE ($R_{\text{CAO}} \approx 0.12-0.20$)
compared to the GSE ($R_{\text{CAO}} \approx 0.23-0.33$). Thus, despite occupying slightly more than 20% of the area, CAOs in the KOE
explain only 20-30% of the total variance.





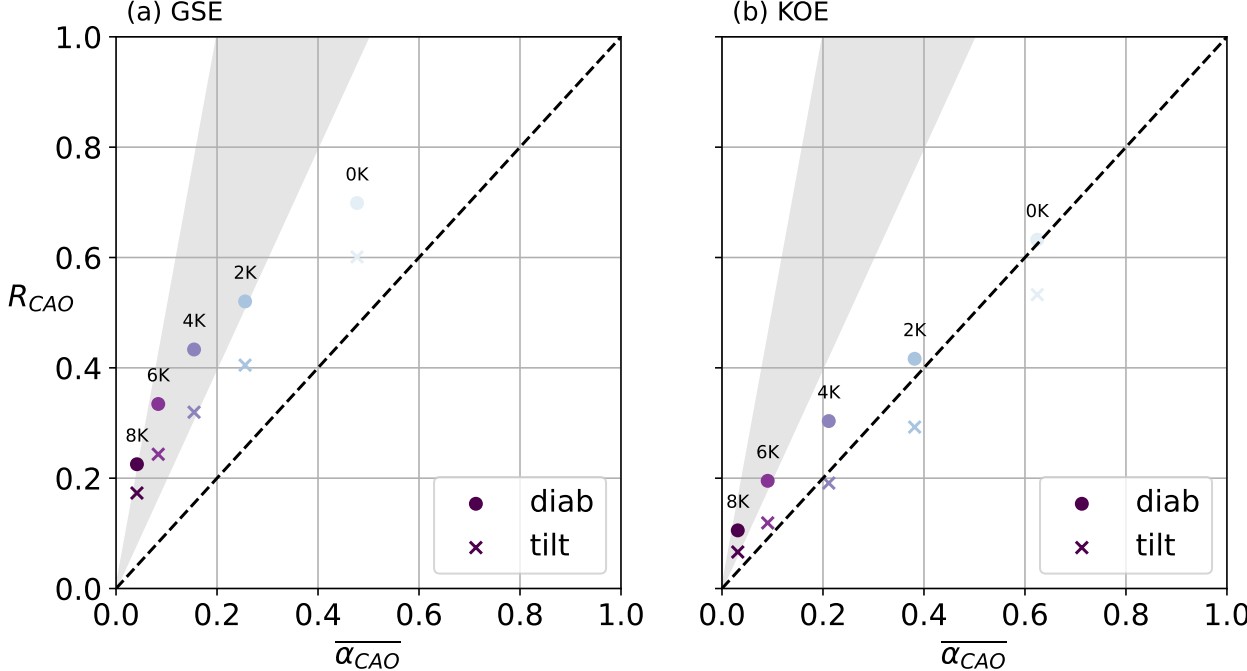

**Figure 2.** Fraction $R_{CAO}$ (Eq. 10) of DIAB (dots) and TILT (crosses) plotted against the mean area fraction occupied by $\overline{\alpha_{CAO}}$. Markers are colour-coded according to the CAO index threshold (Eq. 3), with corresponding values annotated. The dashed line indicates the 1-1 line, while the grey shaded area highlights the sector where the ratio between $R_{CAO}$ and mean $\overline{\alpha_{CAO}}$ is in the range 2:1–3:1.

The strongest CAOs (CAO index$\geq$ 8K) account for two to three times more of the total DIAB variance than their spatial extent would imply. In contrast, for TILT, the share of total variance closely aligns with its areal extent, which suggests TILT variability is not necessarily associated with CAOs. For other CAO categories, variance generally scales with the areal extent of CAOs, while moderate and weak CAOs (CAO index$\leq$ 4K) explain even less of the total variance in TILT than their areal extent (Fig. 2b). Overall, CAOs in the KOE region thus play a different role in the evolution of near-surface baroclinicity compared to the GSE region.

## 4 CAO-based phase portraits

To assess the coherence of variability explained by CAOs with the total variability, we focus on CAOs of moderate and stronger intensity (CAO index$\geq$ 4K; Papritz and Spengler, 2017) and construct phase portraits of DIAB and TILT for the total (DIAB$_{tot}$, TILT$_{tot}$, Fig. 3a,d), CAO (DIAB$_{CAO}$, TILT$_{CAO}$, Fig. 3b,e), and background (DIAB$_{noCAO}$, TILT$_{noCAO}$, Fig. 3c,f) time series.

Phase portraits based on total variability for the GSE (Fig. 3a) and KOE (Fig. 3d) regions are qualitatively similar. The primary circulation is in the anticlockwise direction, indicating that TILT leads in time over DIAB. Thus baroclinicity is first adiabatically depleted by tilting and subsequently restored diabatically, consistent with Marcheggiani and Spengler (2023).







**Figure 3.** Phase portraits of DIAB (x-coordinate) and TILT (y-coordinate) constructed using total (a,d), CAO (b,e), and background (c,f) time series for the GSE (a-c) and KOE (d-f) regions. Kernel-averaged mean-slope is shown in shading, offset and scaled according to the mean and standard deviations of the slope time series, respectively, annotated in the upper right corner of each panel. Contours represent the stream function associated with the kernel-averaged phase space circulation. Points where data is scarce (less than 0.5% of maximum data density) are blanked out. The size of the Gaussian filter used to construct the phase portrait is indicated by the black-shaded dot in the lower-left corner. Star markers indicate locations in the phase space where the kernel composites are evaluated (see Fig. 4).





For both regions, the steepest slope coincides with strongest DIAB and TILT, while the mean slope is larger over the KOE

(4.2 m/km) compared to the GSE (3.4 m/km). The amplitude of the oscillations reaches up to 30 m/km day$^{-1}$ in both regions, though the phase space flow for the KOE region is somewhat noisier, featuring a secondary clockwise circulation when TILT exceeds -30 m/km day$^{-1}$ (Fig. 3d).

Phase portraits based on CAO variability (Fig. 3b,e) are qualitatively identical to the ones for the total variability (Fig. 3a,d). In both the GSE and KOE regions, the primary circulation in the phase space is also anticlockwise, indicating that the lead in

time of TILT over DIAB is strongly linked with CAO-related variability. The secondary clockwise circulation at strong TILT in the KOE region is reproduced for the CAO time series, suggesting it is also linked to CAO dynamics. The magnitude of oscillations in the CAO phase portraits is comparable to those in total variability phase portraits, with values of DIAB reaching up to 20 m/km day$^{-1}$, just slightly weaker than for the total variability (up to 25-30 m/km day$^{-1}$).

In the background, on the other hand, the typical evolution of DIAB and TILT is significantly different. For GSE, DIAB

leads in time over TILT, evident by the clockwise direction of the main phase space circulation (Fig. 3c,f), which actually resembles the typical evolution observed in the free troposphere (Marcheggiani and Spengler, 2023). The circulation is also noisier, as secondary ripples depart from the main circulation. Furthermore, the amplitude of the oscillations is weaker than that for the total and CAO phase space.

The background-based phase portrait for the KOE region exhibits an even higher degree of noise, making it difficult to

identify any clear or consistent circulation pattern (Fig. 3e). Instead, the phase portrait is characterised by numerous smaller-scale circulation cells that appear scattered and disorganised, suggesting that the observed variability may be dominated by noise rather than physical mechanisms.

## 5 Synoptic perspective

To gain further mechanistic insights, we present the synoptic conditions associated with the strongest TILT along the primary

circulation within the total, CAO, and background phase portraits. The background phase portrait for the KOE region is particularly noisy and there is no clear circulation, so we select a point with similar proportions between the magnitudes of TILT and DIAB as in the other phase portraits. The specific location of these points in each phase portrait is marked in Fig. 3.

Phase composites based on the total variability time series for the GSE (Fig. 4a) and KOE (Fig. 4b) regions feature negative, cyclonic geopotential anomalies both at the surface and in the free troposphere. Free tropospheric anomalies are shifted west-

wards compared to surface anomalies, suggesting baroclinic development. This configuration is also conducive to advection of cold air masses from the North American (Fig. 4a) or the East Asian (Fig. 4ab) continents, respectively, which is reminiscent of the typical onset of CAOs in these regions (Kolstad et al., 2009; Fletcher et al., 2016).

The anomalous moisture distribution is also consistent with advection of drier continental air masses over the western North Atlantic and North Pacific oceans, as total column water vapour (TCWV) is up to 15% and 30% lower than climatology over

the East Asian and North American continents, respectively. Additionally, regions of strong TILT extend farther east over both the North Atlantic and North Pacific than regions of strong DIAB.





**Figure 4.** (a,c,e) GSE and (b,d,f) KOE phase composite for (a,b) total, (c,d) CAO, and (e,f) non-CAO phase portraits (Fig. 3). Contours of Z1000 and Z500 (red and blue, respectively) represent the anomaly field relative to climatology (NDJF, 1979–2020) and are plotted every 4 dam (0 dam contours omitted, negative contours dashed). Ratio of TCWV to climatology is shown in black contours (±15% and ±30%, negative values dashed). Grid points were CAO occurrence is above 40% are stippled. Shading for DIAB (red) and TILT (blue) indicates values above 15 m/km day$^{-1}$ for DIAB, below -15 m/km day$^{-1}$ for TILT. Area-mean values of DIAB and TILT are annotated outside of each panel.





Composites at different, adjacent points in the phase space (not shown) reveal that peaks in both DIAB and TILT propagate eastward over the ocean. The fact that TILT peaks further downstream than DIAB is consistent with TILT leading in time over DIAB observed in the phase portraits (Fig. 3a,d).

Phase composites based on CAO time series (Fig. 4c,d) are qualitatively identical to those based on total variability in both the GSE (compare (Fig. 4a,c) and KOE (compare (Fig. 4b,d) regions. Negative geopotential anomalies are slightly stronger than in the total variability composites. Similarly, the dry signal from cold continental air masses is amplified, as areas of 15-30% lower TCWV expand farther inland over North America (Fig. 4c) and East Asia (Fig. 4d). The distribution of CAO occurrence is also qualitatively unchanged between total and CAO phase composites, reflecting the fact that CAOs are the

primary contributors to total variability in both regions.

A significantly different picture emerges from background composites for both GSE and KOE (Fig. 4e,f), where the signal almost disappears. The sign of the geopotential height and TCWV anomalies is actually opposite, indicating weakly anticyclonic flow and slightly higher moisture content (up to 15% more than climatology). Over the North Atlantic (Fig. 4e), both DIAB and TILT are substantially weaker, as is the frequency of CAOs. In contrast, the KOE background composite still shows

relatively strong DIAB, especially near the western North Pacific coast, along with a comparatively high frequency of CAOs.

The persistence of CAOs, even in the background composite, reflects the climatologically frequent occurrence of CAOs and stronger DIAB in KOE, reaching up to 60% of all winter days (not shown). The higher CAO frequency in the KOE is likely driven by the combined influence of the East Asian winter monsoon (Zhang et al., 1997) and the East Asian cold air stream (Compo et al., 1999; Iwasaki et al., 2014), both of which favour frequent and extended cold surges. Consequently, the relatively

high CAO occurrence is partially retained when compositing.

## 6   Conclusions

Cold air outbreaks (CAOs) are a recurrent synoptic feature in the North Atlantic and North Pacific storm track entrance regions. The strong surface heat fluxes associated with CAOs exert a significant influence on the variability of near-surface baroclinicity by anchoring midlatitude storm tracks to the strong SST gradients associated with western boundary currents. To clarify the

role of CAOs in near-surface baroclinicity, we isolated diabatic and adiabatic changes in baroclinicity by using the isentropic slope framework (Papritz and Spengler, 2015), and quantified the contribution of CAOs to the total baroclinicity variability in the near-surface troposphere (900-825 hPa), where baroclinicity is diabatically restored following an initial adiabatic depletion typically associated with the advection of cold air masses (Marcheggiani and Spengler, 2023). While cyclones and fronts constitute the main contributors to the total variability of free-tropospheric baroclinicity (Marcheggiani et al., 2025), our results

highlight the distinct and complementary role of CAOs in driving baroclinicity variability in the lower troposphere.

Despite their relatively small area, CAOs account for most of the total variability in near-surface baroclinicity. In the North Atlantic, CAOs of moderate intensity (CAO index above 4K, see Eq. 2) account for around 40% of the total variability in near-surface baroclinicity while extending over less than 15% of the Gulf Stream Extension (GSE) region (Fig. 2a). In the North Pacific, the contribution from CAOs to total variability is less clear, as they are found to explain less than 30% of the



total variability despite occupying, on average, more than 20% of the Kuroshio-Oyashio Extension (KOE) region (Fig. 2b). The larger areal extent compared to CAOs in the North Atlantic is likely due to the combined effect of the East Asian winter monsoon and East Asian cold air stream (Zhang et al., 1997; Iwasaki et al., 2014), which allows for cold continental air masses associated with CAOs to reach further south-west into the Pacific Ocean compared to CAOs in the North Atlantic.

Most of the adiabatic depletion through tilting of isentropic surfaces (TILT) and diabatic restoration (DIAB) of near-surface
baroclinicity takes place within CAOs. The particular phasing between TILT and DIAB, with TILT leading in time on DIAB, is also primarily associated with CAOs. This is evidenced by phase portraits based on CAO-related baroclinicity variability (Fig. 3b,e), which are qualitatively identical to those based on total variability (Fig. 3a,d) and consistent with previous results by (Marcheggiani and Spengler, 2023). The signature of CAOs is also evident from composites based on strong TILT and DIAB, where the synoptic conditions emerging from both total (Fig. 4a,b) and CAO (Fig. 4c,d) variability feature cyclonic
flow over both GSE and KOE region, advecting cold, dry air off the North American and Northwestern Asian continents over the adjacent ocean basins.

Background, non-CAO based phase portraits (Fig. 3c,f), on the other hand, are significantly different. In the GSE region, the phasing between background DIAB and TILT is reversed and actually more similar to what is observed in the free troposphere (see Marcheggiani and Spengler, 2023, their Fig. 3). The limited influence of background variability on the total variability
provides further evidence to the importance of CAOs in explaining the lower-upper troposphere dichotomy in the maintenance of baroclinicity. Compared to CAO-based composites, the signal emerging from composites of strong background TILT and DIAB (Fig. 4e,f) is significantly weaker and of opposite sign, as only weak anticyclonic and dry anomalies are detected.

The background-based phase space circulation in the KOE (Fig. 3f) is largely incoherent and affected by a higher degree of noise compared to the GSE. This points to the lack of a clear phasing between DIAB and TILT variability in the KOE
background (i.e., outside of CAOs). This is partly explained by the more frequent and deeper south-westward extension of CAOs in the KOE region compared to CAOs in the GSE, as evidenced by the relatively high occurrence of CAOs in the KOE, even when excluding CAOs (Fig. 4f). Since moderate-intensity CAOs appear to develop continuously over the KOE, background variability becomes marginal. CAOs in the KOE thus play a more dominant role in shaping near-surface baroclinicity variability than in the GSE.

Given their central role in restoring near-surface baroclinicity, our results indicate that CAOs should be considered an integral part of the life cycle of midlatitude storm tracks. Recognising the role of CAOs in restoring near-surface baroclinicity not only highlights their contribution to storm track maintenance, but also underscores their relevance for advancing our theoretical understanding of how diabatic processes near the surface shape storm track evolution and their interaction with oceanic boundary currents.

*Code availability.* The Python library *Dynlib* (Spensberger, 2024) is freely available at https://doi.org/10.5281/zenodo.10471187, and ERA5 data (Hersbach et al., 2020) are freely available at https://www.ecmwf.int/en/forecasts/dataset/ecmwf-reanalysis-v5.





*Data availability.* ERA5 data is freely available from the ECMWF at https://www.ecmwf.int/en/forecasts/datasets/reanalysis-datasets/era5.

*Author contributions.* AM performed data analyses and prepared the paper. TS contributed to the interpretation of the results and to the writing of the paper.

*Competing interests.* The contact author has declared that neither of the authors has any competing interests.

*Acknowledgements.* This work was funded by the Research Council of Norway (NFR) via the BALMCAST (324081) and ARCLINK (328938) projects.



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
