# Peer review of "Cold air outbreaks drive near-surface baroclinicity variability over storm track entrance regions in the Northern Hemisphere"

_EGUsphere, 2025_

## Referee Comment (RC1)

Review of: Cold air outbreaks drive near-surface baroclinicity variability

Authors: Andrea Marcheggiani and Thomas Spengler

This paper investigates the role of CAOs for variability in near-surface baroclinicity at the entrance of the Northern Hemisphere storm tracks, along the Kuroshio-Oyashio and Gulf Stream Extensions, from the perspective of the isentropic slope framework.

The authors quantify the fraction of total variability in lower tropospheric baroclinicity (measured in the variability in the isentropic slope) explained by various intensities of CAOs (measured using a CAO index). They find that a substantial fraction of variability in the slope in the Gulf Stream region can be attributed to CAOs, whereby particularly strong CAOs account for a disproportionally fraction of variability in baroclinicity. For the Kuroshio-Oyashio Extension CAOs appear to explain a smaller fraction in total variability of baroclinicity, leading the authors to the conclusion, that the role of CAOs for determining baroclinicity in the storm tracks differs between the two study regions.

The authors complement these results with an analysis of the two contributing terms (tilting and diabatic term) in a phase space - to showcase how the two terms evolve in time, and contrast the CAO's contributions with the background contribution – as well as a synoptic perspective providing more context as to what the situations look like when the tilting term is at its maximum.

Overall, the study is innovative and sheds new light onto an important topic of the midlatitudes: The relevance of CAOs for the downstream storm tracks. The manuscript is carefully prepared, very-well written and features nicely prepared figures. I also enjoyed the synoptic perspective section, which can offer more concrete context to the more abstract phase spaces.

Main comments:

1. The isentropic slope framework, as I understand, quantifies baroclinicity as a means of the slope of isentropes. While the isentropes are slanted in the free troposphere, they are vertical in the convectively mixed boundary layers and may even be unstable in extreme situations such as marine CAOs (Vannière et al. 2017). During moderate to intense CAO events the marine boundary layer may well reach 825hPa, which is here chosen as the upper boundary. Therefore, I would be curious to know how this issue is treated in this study, and how grid points in which $\theta$ is constant throughout the column from 900-825hPa are handled.

2. Related to main comment 1, I'm also wondering about the representation of potential temperature in the marine boundary layer in ERA5. How well can we trust the profile of potential temperature within the boundary layer (especially in extreme situations such as marine CAOs)?

3. CAO indices smaller than 4K can hardly be considered CAOs. Though, it is very interesting to see by how much various intensities of CAOs contribute to variability, I think it is misleading to call regions with, e.g., CAO index > 2K, CAOs, as these regions cover a large fraction of the study region (see Fig. 1) and are often likely associated with cold sectors of cyclones rather than transport of cold air from the polar regions/cold continent associated with CAOs. Consider discussing this issue more explicitly for CAO indices smaller than 4K.

4. I understand that you exclude the Japan Sea, as this region is not directly connected to the storm tracks, yet this region features a lot of very strong CAOs (stronger than on the eastern side of Japan) and large baroclinicity due to the strong land-sea contrast. Furthermore, large baroclinicity in this region may still be important for cyclones developing at the entrance of the storm tracks. How would your results change if you included this basin? Would a larger fraction of the variability in baroclinicity be explained by CAOs then?

Specific comments:

- Line 40: Please add a space between troposphere and the parenthesis.

- Figure 1 caption: "23 January 2014 at 15Z". Do you mean 15 UTC?

- Line 62-63: I understand that you exclude the Japan Sea, as this region is not directly connected to the storm tracks, yet this region features a lot of very strong CAOs (stronger than on the eastern side of Japan) and large baroclinicity due to the strong land-sea contrast. Furthermore, large baroclinicity in this region may still be important for cyclones developing at the entrance of the storm tracks. How would your results change if you included this basin? Would a larger fraction of the variability in baroclinicity be explained by CAOs then?

- Line 74: Suggest change to "ADV represents adiabatic changes in the slope …"

- Line 75-76: If the magnitude of the advection term is comparable to the other two terms, a substantial fraction and potential source in variability of baroclinicity is neglected. How would this term be related to/affected by the occurrence of CAOs?

- Line 141 -146: I would argue that a large part of this larger extent is not the CAO itself but rather the remnants of a cold sector of a cyclone over warmer ocean surfaces (see main comment 2). From Fig. 1 it is evident that it is mainly the extent of the region where the CAO index < 4K, differs between the 2 regions. The extent of CAO index > 6K (as well as CAO index > 8K) seems to me of comparable size in the two basins. The current interpretation may be misleading to other readers. Consider discussing this matter more explicitly.

- Line 138-140: Tilting term → how trustworthy/significant is this is, as you say, a very noisy field, featuring a lot of small-scale dipoles due to variability of up and downdrafts within the Boundary Layer, and then average over a large spatial domain → don't we just get a residual of large values in dipoles?

- Line 152: Consider adding a reference to Fig. 2 after "than their spatial extent would imply".

- Line 155: I find this mismatch between the explained variability between the two regions very interesting. Do you have any explanation why this may be the case?

- Line 162: Add a comma after "Thus,…".

- Line 162-163: where is the starting point in the analysis of phase diagrams? How can one start by saying that "first" baroclinicity is depleted by tilting and "then" diabatically restored? Maybe this is explained in the cited reference, but a quick hint in this direction would also be beneficial to this manuscript.

- Line 170: Why is there a secondary circulation with a clockwise direction? Also, how would this exactly be explained by CAO dynamics, as stated in Line 171?

- Lines 182: what would this "noise" represent physically?

- Line 188: The described sentence appears to be different from what is shown in Fig. 4. Specifically, the authors speak of the presence of a cyclone indicated by the contour lines, yet from the figure caption I take that solid contours indicate positive anomalies (and thus an anticyclone).

- Figure 4: Why are there faint and solid blue and red lines? Also, how significant are these anomalies?

- Line 201-202: Same as in Line 188.

- Line 211: I'd rather say, there is persistent advection of cold continental air, resulting in climatologically higher CAO indices. Whether this is actually a CAO or not is debatable….

---

## Author Comment (AC1)

**Cold air outbreaks drive near-surface baroclinicity variability**

**Andrea Marcheggiani and Thomas Spengler**

Geophysical Institute, University of Bergen, and Bjerknes Centre for Climate Research, Bergen, Norway

October 3, 2025

**Authors response to reviewers**

We are thankful for the constructive comments from all the reviewers. We hope that all their concerns have been duly addressed in the revised version of this paper.

Comments by the reviewers are in **bold**, followed by our replies. Figures from the original manuscript are referred to following the manuscript's order while new figures included in this document are labelled as Figure AR# (Author Response).

**Response to Reviewer 1**

This paper investigates the role of CAOs for variability in near-surface baroclinicity at the entrance of the Northern Hemisphere storm tracks, along the Kuroshio-Oyashio and Gulf Stream Extensions, from the perspective of the isentropic slope framework. The authors quantify the fraction of total variability in lower tropospheric baroclinicity (measured in the variability in the isentropic slope) explained by various intensities of CAOs (measured using a CAO index). They find that a substantial fraction of variability in the slope in the Gulf Stream region can be attributed to CAOs, whereby particularly strong CAOs account for a disproportionally fraction of variability in baroclinicity. For the Kuroshio-Oyashio Extension CAOs appear to explain a smaller fraction in total variability of baroclinicity, leading the authors to the conclusion, that the role of CAOs for determining baroclinicity in the storm tracks differs between the two study regions. The authors complement these results with an analysis of the two contributing terms (tilting and diabatic term) in a phase space – to showcase how the two terms evolve in time, and contrast the CAO's contributions with the background contribution – as well as a synoptic perspective providing more context as to what the situations look like when the tilting term is at its maximum.

Overall, the study is innovative and sheds new light onto an important topic of the midlatitudes: The relevance of CAOs for the downstream storm tracks. The manuscript is carefully prepared, very-well written and features nicely prepared figures. I also enjoyed the synoptic perspective section, which can offer more concrete context to the more abstract phase spaces.

We thank the Reviewer for the positive and constructive review of our work. We hope to have addressed all the concerns raised. Below we provide a line-by-line response.

**Main comments**

1) The isentropic slope framework, as I understand, quantifies baroclinicity as a means of the slope of isentropes. While the isentropes are slanted in the free troposphere, they are vertical in the convectively mixed boundary layers and may even be unstable in extreme situations such as marine CAOs (Vannière et al. 2017). During

moderate to intense CAO events the marine boundary layer may well reach 825hPa, which is here chosen as the upper boundary. Therefore, I would be curious to know how this issue is treated in this study, and how grid points in which  $\theta$  is constant throughout the column from 900-825hPa are handled.

As pointed out by the Reviewer, turbulent mixing within the boundary layer leads to nearly vertical isentropic surfaces and thus especially low static stability  $\partial\theta/\partial z$ . To avoid any numerical issues related to extremely steep isentropic slopes, we follow Papritz and Spengler (2015) and mask out any grid point where  $\partial\theta/\partial z < 10^{-4} {\rm K~m^{-1}}$ . Therefore, in those cases where the boundary layer is particularly deep, we only consider grid points that do not directly fall into it, focusing on the dynamics above. As this was not explicitly mentioned in the original manuscript, we have expanded lines 84-85 to clarify our methods.

2) Related to main comment 1, I'm also wondering about the representation of potential temperature in the marine boundary layer in ERA5. How well can we trust the profile of potential temperature within the boundary layer (especially in extreme situations such as marine CAOs)?

The reviewer raises a valid concern, which we had also considered during the early stages of this work. To address this, we examined radiosonde data from three stations located in the North Sea and Norwegian Sea, provided by the Norwegian Meteorological Institute, and compared it with ERA5 data for the period 2000–2009. This comparison revealed no substantial differences, consistent with the findings of Seethala et al. (2021), who demonstrated that ERA5 biases do not significantly compromise its reliability, particularly during winter. Further supporting this, Lavers et al. (2020) found that biases in temperature, wind, and humidity in ERA5 are most pronounced below 900 hPa (levels not included in our analysis) and even then, the biases remain relatively small (e.g., 0.6 K for temperature). Additionally, Simmons (2022) compared ERA5 with other reanalysis (JRA-55) and observational products and reported overall agreement, with only minor regional discrepancies in areas not relevant to our study. Moreover, our methodology implicitly excludes grid points within the mixed layer of the boundary layer, where static stability is low. As our focus is on baroclinicity variability just above the boundary layer, we do not expect ERA5 model biases to meaningfully affect our results. We have expanded the manuscript (lines 87-89) to include this discussion.

3) CAO indices smaller than 4K can hardly be considered CAOs. Though, it is very interesting to see by how much various intensities of CAOs contribute to variability, I think it is misleading to call regions with, e.g., CAO index > 2K, CAOs, as these regions cover a large fraction of the study region (see Fig. 1) and are often likely associated with cold sectors of cyclones rather than transport of cold air from the polar regions/cold continent associated with CAOs. Consider discussing this issue more explicitly for CAO indices smaller than 4K.

A threshold of 4K is commonly used to identify CAOs and to distinguish them from general cold air advection. Our aim, however, was to assess the sensitivity of our analysis to CAOs of varying intensity and spatial extent. To this end, we adopted the categorisation proposed by Papritz and Spengler (2017), which defines CAO indices between 0K and 4K as weak, 4K to 8K as moderate, and so on. We should also point out that by lowering the threshold to 2K, we do not exclude stronger CAOs; rather, we expand the range to include both weak and more intense events. We expanded the original manuscript on lines 96-101 to include this discussion.

4) I understand that you exclude the Japan Sea, as this region is not directly connected to the storm tracks, yet this region features a lot of very strong CAOs (stronger than on the eastern side of Japan) and large baroclinicity due to the strong land-sea contrast. Furthermore, large baroclinicity in this region may still be important for cyclones developing at the entrance of the storm tracks. How would your results change if you included this basin? Would a larger fraction of the variability in baroclinicity be explained by CAOs then?

When focusing on the Sea of Japan alone, we observe a phase space circulation pattern similar to that found in the Gulf Stream and Kuroshio-Oyashio

regions, namely, anticlockwise oscillations in the negative-TILT, positive-DIAB quadrant. These oscillations exhibit larger amplitudes, likely due to the smaller spatial extent of the averaging domain. However, the variability in the Sea of Japan does not appear to be coherent with that of the Kuroshio-Oyashio Extension, as the corresponding phase portraits for both regions are noticeably noisier. For this reason, we chose to exclude the Sea of Japan from the spatial averaging and concentrate on the main North Pacific storm track entrance region. We have included this discussion on lines 139-146 in the Methods section.

**Minor comments**

• Line 40: Please add a space between troposphere and the parenthesis.

Thanks for spotting the typo, this is fixed now.

• Figure 1 caption: "23 January 2014 at 15Z". Do you mean 15 UTC?

Indeed, we have changed "15Z" to "1500 UTC" for clarity.

• Line 62-63: I understand that you exclude the Japan Sea, as this region is not directly connected to the storm tracks, yet this region features a lot of very strong CAOs (stronger than on the eastern side of Japan) and large baroclinicity due to the strong land-sea contrast. Furthermore, large baroclinicity in this region may still be important for cyclones developing at the entrance of the storm tracks. How would your results change if you included this basin? Would a larger fraction of the variability in baroclinicity be explained by CAOs then?

See response to main comment 4 above.

• Line 74: Suggest change to "ADV represents adiabatic changes in the slope ..."

We have rephrased the entire paragraph and provided a more detailed clarification of the role of the ADV term (lines 75-81).

• Line 75-76: If the magnitude of the advection term is comparable to the other two terms, a substantial fraction and potential source in variability of baroclinicity is neglected. How would this term be related to/affected by the occurrence of CAOs?

The advection term is indeed associated with a substantial fraction of baroclinicity variability. However, the physical interpretation of this term is not so clear, as it primarily represents the advection of slope by the atmospheric flow. It is not associated with any specific physical mechanism that modifies isentropic surfaces. Thus, we excluded it from our analysis to focus on the tilting term, which has a direct physical interpretation as the mechanical tilting of isentropic surfaces by the atmospheric flow. In the specific case of CAOs, the advection term would intensify as the cold air masses move into the Gulf Stream and Kuroshio-Oyashio Extension regions. We have expanded lines 75-76 of the original manuscript to include this discussion (lines 75-81 in the revised manuscript).

• Line 141 -146: I would argue that a large part of this larger extent is not the CAO itself but rather the remnants of a cold sector of a cyclone over warmer ocean surfaces (see main comment 2). From Fig. 1 it is evident that it is mainly the extent of the region where the CAO index < 4K, differs between the 2 regions. The extent of CAO index > 6K (as well as CAO index > 8K) seems to me of comparable size in the two basins. The current interpretation may be misleading to other readers. Consider discussing this matter more explicitly.

It is true that the CAO index likely captures more than just canonical cold air outbreaks, particularly when using thresholds below 4K, which tend to encompass a broader area in the North Pacific compared to the North Atlantic.

That said, 4K is a commonly used threshold for identifying CAOs and, from a thermodynamic perspective, should reflect similarly strong air—sea interactions. Distinguishing between CAOs and the cold sectors of extratropical cyclones is inherently complex and falls outside the scope of this study. However, we have expanded lines 96-101 to avoid any confusion and clarify that the index may capture more than CAOs in the North Pacific.

• Line 138-140: Tilting term -> how trustworthy/significant is this is, as you say, a very noisy field, featuring a lot of small-scale dipoles due to variability of up and downdrafts within the Boundary Layer, and then average over a large spatial domain -> don't we just get a residual of large values in dipoles?

We acknowledge that the tilting term exhibits a noisy structure, often characterised by small-scale dipoles resulting from the variability of up- and downdraughts within the boundary layer. While this spatial variability is indeed significant, the field is averaged over a large domain, effectively smoothing out much of the local noise. What remains is a consistent negative mean throughout the entire climatological period, which we interpret as indicative of the net negative effect of tilting on the isentropic slope. So, although the field contains dipoles, the averaging does not simply leave behind residuals of large values, but it reflects a coherent signal that supports our interpretation.

• Line 152: Consider adding a reference to Fig. 2 after "than their spatial extent would imply".

We have included a reference to Figure 2.

• Line 155: I find this mismatch between the explained variability between the two regions very interesting. Do you have any explanation why this may be the case?

As the Reviewer pointed out in a previous comment, similar CAO index thresholds may detect quite different synoptic events across the two basins, so that the CAO mask used in the North Pacific actually encompasses more than CAOs, including unrelated variability. As the CAO index threshold is relaxed, larger areas are included in the spatial average, and the ratio of total variance explained,  $R_{CAO}$ , increases proportionally with the area fraction. We therefore speculate that the observed differences between the two ocean basins may stem from the distinct characteristics of CAOs in each region. We have included this discussion on lines 171-174.

• Line 162: Add a comma after "Thus,...".

We added a comma.

• Line 162-163: where is the starting point in the analysis of phase diagrams? How can one start by saying that "first" baroclinicity is depleted by tilting and "then" diabatically restored? Maybe this is explained in the cited reference, but a quick hint in this direction would also be beneficial to this manuscript.

While the choice of a starting point is arbitrary, we decided to start from the point of weakest DIAB and TILT. The lead of TILT over DIAB, on the other hand, is indicated by anticlockwise direction of the phase space circulation: as we follow one of the close trajectories traced by the stream function, we find that the (absolute) increase in TILT is followed by the increase in DIAB, which peaks while TILT already starts decreasing. We have rephrased lines 192-193 to clarify this point.

• Line 170: Why is there a secondary circulation with a clockwise direction? Also, how would this exactly be explained by CAO dynamics, as stated in Line 171?

The secondary, clockwise circulation is associated with particularly strong values of TILT and reflects a genuine mean behaviour in the DIAB-TILT relationship in the Kuroshio-Oyashio Extension. We suspect that the extreme values of TILT are the result of a deep boundary layer extending into

the near-surface troposphere, as suggested by composites of mean boundary layer height (not shown). Although we apply a mask for extreme slope and slope tendencies, large values may still persist near these masked grid points, especially within CAOs, whose phase portrait (Fig. 3e) feature an even stronger secondary circulation. Upon examining the synoptic conditions associated with these secondary circulations, we, however, found no consistent or recurring patterns that would suggest an underlying physical mechanism, hence we focused on the main circulation only. We acknowledge that Line 171 may have been incomplete or unclear, and we have revised the corresponding paragraph in the manuscript to include this discussion (lines 203-211).

**• Lines 182: what would this "noise" represent physically?**

The noise in Fig. 3f is indicative of the absence of any single physical mechanism that can explain the co-variability of DIAB and TILT outside of CAOs in the KOE region. We speculate that whatever drives the variability of TILT and DIAB outside of CAOs is not well organised in space nor in time, which makes it particularly challenging to capture in the synoptic composites, where they may be lost during the averaging process. We have expanded line 182 to include this clarification (lines 219-223).

• Line 188: The described sentence appears to be different from what is shown in Fig. 4. Specifically, the authors speak of the presence of a cyclone indicated by the contour lines, yet from the figure caption I take that solid contours indicate positive anomalies (and thus an anticyclone).

We apologise for the confusion, the text in the caption was actually wrong as it was referring to a previous version of the same figure where negative anomalies were represented by dashed contours and positive anomalies by solid contours. As the readability of the contours was not ideal when dashed contours superimpose, we decided to use thick and thin contours to represent negative and positive anomalies, respectively. Therefore, the sentence on line 188 of the original manuscript is actually consistent with Fig. 4. We have fixed the caption in the revised manuscript.

• Figure 4: Why are there faint and solid blue and red lines? Also, how significant are these anomalies?

Again, apologies for the confusion, the caption was referring to a previous version of the same figure where negative geopotential anomaly contours were marked by dashed lines. In the final version, negative, cyclonic anomalies are highlighted by thick solid lines, while the thin solid lines represent positive, anticyclonic anomalies. As for the significance of the anomalies, these emerge from averaging over 42 winters and are relatively strong compared to cases where there is no evident signal, as we find for background composites. For this reason we trust these anomalies to carry enough physical significance base our interpretation of the phase space circulation.

• Line 201-202: Same as in Line 188.

See response provided above.

• Line 211: I'd rather say, there is persistent advection of cold continental air, resulting in climatologically higher CAO indices. Whether this is actually a CAO or not is debatable....

We appreciate the Reviewer's perspective and agree that the CAO index is not a perfect identifier for CAOs. Nonetheless, it is widely used in the literature to define CAOs, and assessing the accuracy or skill of such identification methods falls outside the scope of this study. We also suggest that the larger and more frequent CAOs observed in the KOE region may be driven by stronger cold air advection from the continent. Whether this advection differs fundamentally from canonical CAOs is an open question, which we leave for future work to explore.

**Response to Reviewer N.2**

The authors present an analysis of the role that cold air outbreaks play in defining the baroclinicity over oceanic mid-latitudes (the North Atlantic and North Pacific basin). They analyse winters from a 40-year period using ERA5 reanalyses. They employ the relatively new and still novel isentropic slope framework to quantify the contribution to baroclinicity, finding that CAO play a substantial role in the Gulf Stream and Kuroshio regions. The key metric that supports this finding is the radio of CAO variance (normalised by the total variance) for the DIAB and TILT diagnostics, which can reach 40-45% for the Gulf Steam, even though the area of CAO is half that amount.

Overall, this is a concise, well written and nicely illustrated article. I don't have any major points to address. However, I would say the article asks quite a lot of its readers. Those not so familiar with the isentropic slope framework don't get a lot of help with interpretation here. This study builds on several previous ones outlining this framework and I wonder if a little more explanation would be beneficial here.

We thank the Reviewer for their positive review of our paper. We understand the main concern raised with regards to accessibility of our work to those not so familiar with the isentropic slope framework. In the revised manuscript, we have expanded the method section to provide a clearer interpretation of the various components of the framework and the implications for this study in particular. We hope to have addressed all the concerns raised by the Reviewer.

**Specific comments**

1) I think the title needs a geographic qualifier. The authors have only examined two specific regions and I'd suggest that is reflected in the title. For example, "Cold air outbreaks drive near-surface baroclinicity variability over oceanic mid-latitudes" or "...over western boundary currents".

The Reviewer raises a fair point. While it is true our study is based on two regions that feature western boundary currents, our focus is actually on the corresponding North Pacific and North Atlantic storm tracks, thus we changed the title to "Cold air outbreaks drive near-surface baroclinicity variability over storm track entrance regions in the Northern Hemisphere".

2) Figure 1 – Panel (b) is a smorgasbord of greys. The continents, sea-ice areas and CAO masks are all different shades of grey! I'd recommend changing the CAO masks to the same shade of blue as the 4 K shading in panel (a).

We followed the Reviewer's suggestion and now identify the CAO mask in panel (b) with the same purple shading used for 4-6K in panel (a).

3) Figure 1 – would it be useful to include the TILT term as panel (d) in this figure? It might give opportunity to explain these terms in a bit more detail for readers less familiar with this framework.

We have included a fourth panel with instantaneous values of TILT as suggested by the Reviewer.

4) The ADV advection component of the slope tendency equation is neglected in this study. The reasoning behind this needs more explanation. The one given is "the interpretation is less straightforward". But I don't think that is very satisfactory. Perhaps you can take recourse to previous work where this ADV term is shown to be less important in interpretation? What are the ramifications for neglecting this term?

The other Reviewer raised a similar point, and we agree a clearer motivation for our choice is needed, though the same reasoning motivated a similar choice in Marcheggiani and Spengler (2023) and Marcheggiani et al. (2025). We exclude the ADV term as it represents changes in slope associated entirely with advection by the flow and, as such, not necessarily associated with any specific physical mechanism. The TILT term represents the actual kinematic flattening of isentropic surfaces through baroclinic rearrangement of mass. We included a clearer explanation of our choice on lines 76-81.

5) I'd note that equations (7) and (8) could easily be merged.

We agree and removed equation 7, which was already included in equation 8.

6) I wonder if you could add a sentence or two more explanation for Figure 3 which shows the phase portraits. I wasn't quite sure how to interpret these and also wasn't clear how the slope shaded contours were plotted, i.e., how is the area of where they are plotted determined?

We see the point raised by the Reviewer and have accordingly expanded the methods section related to phase space analysis on lines 134-138, as we thought it to be the most appropriate place to clarify this point. However, we would still refer the reader to previous work where further technical details are provided (e.g., Novak et al., 2017; Marcheggiani et al., 2022, specifically in their appendix A).

7) Figure 4 – I think the caption is incorrect here – the plots seem to have negative anomalies as thick contours and positive anomalies as thin contours? The caption talks about dashed contours.

We thank the Reviewer for spotting this inconsistency, which also confused Reviewer 1. The caption was referring to a previous version of the same figure where negative anomalies were plotted in dashed contours. However, the superposition of dashed contours was hard to read, so we eventually opted for thick and thin contours to represent negative and positive anomalies, respectively. We have fixed the caption accordingly.

**References**

- Lavers, D. A., Ingleby, N. B., Subramanian, A. C., Richardson, D. S., Ralph, F. M., Doyle, J. D., Reynolds, C. A., Torn, R. D., Rodwell, M. J., Tallapragada, V., et al.: Forecast errors and uncertainties in atmospheric rivers, Weather and Forecasting, 35, 1447–1458, 2020.
- Marcheggiani, A. and Spengler, T.: Diabatic effects on the evolution of storm tracks, Weather and Climate Dynamics, 4, 927–942, 2023.
- Marcheggiani, A., Ambaum, M. H., and Messori, G.: The life cycle of meridional heat flux peaks, Quarterly Journal of the Royal Meteorological Society, 148, 1113–1126, 2022.
- Marcheggiani, A., Dacre, H., Spensberger, C., and Spengler, T.: Weather features drive free-tropospheric baroclinicity variability in the North Atlantic storm tracks, Quarterly Journal of the Royal Meteorological Society, in press (accepted), https://doi.org/10.1002/qj.5061, 2025.
- Novak, L., Ambaum, M., and Tailleux, R.: Marginal stability and predator—prey behaviour within storm tracks, Quarterly Journal of the Royal Meteorological Society, 143, 1421–1433, 2017.
- Papritz, L. and Spengler, T.: Analysis of the slope of isentropic surfaces and its tendencies over the North Atlantic, Quarterly Journal of the Royal Meteorological Society, 141, 3226–3238, 2015.
- Papritz, L. and Spengler, T.: A Lagrangian climatology of wintertime cold air outbreaks in the Irminger and Nordic Seas and their role in shaping air—sea heat fluxes, Journal of Climate, 30, 2717–2737, 2017.
- Seethala, C., Zuidema, P., Edson, J., Brunke, M., Chen, G., Li, X.-Y., Painemal, D., Robinson, C., Shingler, T., Shook, M., et al.: On assessing ERA5 and MERRA2 representations of cold-air outbreaks across the Gulf Stream, Geophysical Research Letters, 48, e2021GL094364, 2021.
- Simmons, A. J.: Trends in the tropospheric general circulation from 1979 to 2022, Weather and Climate Dynamics, 3, 777–809, 2022.